

# Field evaluation of a novel charge transfer ionization TOF MS for ambient VOC measurements

Olga Zografou[1], Christos Kaltsonoudis[2], Maria Gini[1], Angeliki Matrali[2], Elias Panagiotopoulos[4], Alexandros Lekkas[4], Dimitris Papanastasiou[4], Spyros N. Pandis[2,3] and Konstantinos Eleftheriadis[1]

[1]Institute of Nuclear & Radiological Science & Technology, Energy & Safety, N.C.S.R. "Demokritos", 15341 Athens, Greece;
[2]Institute of Chemical Engineering Sciences, ICE-HT, Patras, 26500, Greece;
[3]Department of Chemical Engineering, University of Patras, Patras, 26504, Greece
[4]Fasmatech Science & Technology SA, NCSR Demokritos, 15310, Athens, Greece

*Correspondence to*: Konstantinos Eleftheriadis ([elefther@ipta.demokritos.gr](elefther@ipta.demokritos.gr)), Olga Zografou
(o.zografou@ipta.demokritos.gr)

**Abstract.** A newly developed charge transfer orthogonal Time-of-Flight Mass Spectrometer (oToF-MS), was deployed for the first time in the field in order to evaluate its ability to perform online real-time measurements of volatile organic compounds (VOCs) for a prolonged time period. The study focused on urban air sampling and targeted a specific range of VOCs, namely: acetone, isoprene, benzene, toluene, and xylene. The measurement campaign took place from May to August 2023 at the
suburban area in Athens Greece. The measured VOC level were consistent with those reported in previous summer campaigns suggesting that the instrument did not face any unexpected problems moving from the laboratory to the field. The variability of the measurements of the various VOCs was used to gain insights about their sources. Transportation emissions were a dominant source of the BTX compounds (benzene, toluene and xylene). Acetone and isoprene were emitted by both anthropogenic and biogenic sources. During the summer biogenic sources were responsible for most of the isoprene in the site.

## 1 Introduction

Real-time monitoring of atmospheric constituents provides valuable information for the development of air quality management strategies and evaluating the impact of various pollutants on human health and the environment (Laj et al., 2009, Goldstein and Gallbally 2007, Kampa and Castanas, 2008, Yang et al., 2009, Kim et al., 2015). Continuous high-temporal measurements of volatile organic compounds (VOCs) are especially challenging due to the sear number of chemical species involved and their trace levels concentrations High temporal resolution measurements can help in source apportionment of the measured VOCs though appropriate techniques such as Positive Matrix Factorization (PMF) (Kaltsonoudis et al., 2016).

VOCs can be biogenic (bVOCs) or anthropogenic (aVOCs). bVOCs are mainly terpenoids (isoprene, monoterpenes and sesquiterpenes), alkanes, alkenes etc. and are emitted from sources such as trees, the ocean, volcanoes, and soils (Guenther et al., 1995, Atkinson 2000). aVOCs are primarily from vehicular, industrial emissions, and biomass burning and consist of
aromatic compounds, alkanes, alkenes, etc. (Li 2021, Wang 2012, Languille 2020). Oxygenated VOCs (oVOCs), such as alcohols, carbonyl compounds, and organic acids can be both emitted by specific sources or be the products of atmospheric oxidation of primary VOCs. VOCs and oVOCs participate in the production of ozone, secondary organic aerosol formation





and new particle formation (Mellouki et al., 2015, Monks et al., 2015; Hallquist et al., 2009). Ambient particles impact climate by absorbing and scattering solar radiation or by serving as cloud condensation nuclei (CCN) (Rosenfeld 2008), and pose

health risks to humans (Daellenbach et al., 2020).

Measurements of VOCs have been widely performed by Proton Transfer Mass Spectrometers (PTR-MS) due to their low detection limits and their ability to perform long-term, real-time, high resolution measurements (Blake et al., 2009; Yuan, et al., 2017). The principle of PTR-MS operation involves the soft plasma ionization of the sample through proton transfer from water molecules (Lindinger et al., 1998). The advantage of this technique lies in the minimal fragmentation of the parent ion

compared to other techniques, such as electron impact ionization, that produce a great number of fragments (Schwoebel et al., 2011, Tani et al., 2022). The PTR-ToF-MS systems offer high mass resolution and sensitivity, while the Q-PTR-MS systems are known for their precision (Warneke et al., 2001). The existing PTR-MS systems can be further improved by taking advantage of the development of new MS systems, optimizing their performance for selected types of oVOCs and VOCs, etc. Previous VOC monitoring studies in Athens include a Q-PTR-MS (Ionicon) summer campaign at the Demokritos suburban

monitoring station in Agia Paraskevi, Athens (DEM) station in July 2012 and a winter campaign in the center of Athens (National Observatory of Athens) in January and February 2013 by Kaltsonoudis et al. (2016). In these studies, the PTR-MS dataset was analyzed using PMF. Five factors were retrieved for the VOCs in Athens during summer; a traffic-related factor, a factor with monoterpene species, one related to other biogenic emissions and two oVOC factors. Panopoulou et al (2020) focused on the concentration of isoprene, limonene and a-pinene using two gas chromatographs with flame ionization detectors

deployed at the National Observatory of Athens to measure the C2-C6 and C6-C12 VOCs (Panopoulou et al., 2020).

In Kaltsonoudis et al. (2023), the newly developed charge transfer oToF-MS was tested under laboratory conditions for VOCs and Intermediate VOCs (IVOCs) measurements. The apparatus consists of an inlet interface, an ionization chamber, an RF ion transfer line, and a Reflectron Time-of-Flight Mass Analyzer. Under experimental conditions the instrument performed measurements of different organic compounds (including aromatics, terpenes, oxygenated VOCs and aliphatic compounds)

with minimum fragmentation and very low detection limits were observed, even in the range of low ppt. Both proton transfer (by oxonium ions) and charge transfer (in the presence of $NO^+$ and $NO_2^+$) reactions were found to ionize the sampled organic molecules. Among the remarkable features of the instrument, its transverse ionization source arrangement enables increased sampling flow rates for enhanced sensitivity. The mass resolving power of the orthogonal ToF at m/z 250 Th was 15000 (fwhm).

The current work aims to evaluate the capabilities of this newly developed ToF-MS system that combines high sensitivity and mass resolution, and low detection limits for atmospheric VOC measurements. Field deployment results from ambient VOCs measurements at a suburban station are presented, focusing on the following compounds: acetone, isoprene, benzene, toluene, and xylene. The measurements were conducted at the DEM station, from May to September 2023, and were complemented by meteorological and air quality measurements from collocated instruments.



## 2 Materials and methods

### 2.1 Novel charge transfer oToF-MS

Details about the new charge transfer oToF-MS for the detection of ambient VOCs can be found in Kaltsonoudis et al. (2023). In summary the main components are the inlet interface, the ionization chamber, the RF ion transfer line, and the Reflectron Time-of-Flight Mass Analyzer.

The inlet interface includes two similar capillary sample inlets in a vertical arrangement; one is used for introducing the gas sample and the other one is auxiliary. Heated (up to 100 ºC) stainless steel 1/8 tubing with a stainless-steel needle valve is used followed by a heated capillary of 70 μm inner diameter that can be set to temperatures up to 150 ºC. A stainless steel filter holder that hosts a 0.22 μm PTFE filter to prevent particles from entering is used prior to the heated sections. Within the interface, a separate inlet is used to introduce humidified nitrogen (pure nitrogen that has passed through a stainless-steel vessel filled with high purity liquid chromatography (HPLC) water. The humidified nitrogen flow is used to provide the necessary water vapors for the production of hydronium ions $(H_2O)H^+$ inside a ceramic tube that contains three ring electrodes of different voltages.

The ionization chamber, merges the gas sample flow and the excited humidified nitrogen in order to ionize the sample analytes. The partial pressures are set to 1.1 mbar for the sample inlet and 1.3 mbar for the nitrogen flow, resulting in a total pressure of 2.4 mbar inside the ionization chamber (stable glow discharge can be achieved in the range of 1-5 mbar). The dominant form of ionization under these conditions is proton transfer or electron removal with a sample analyte being detected at a mass to charge ratio (m/z) of plus one in respect to the molecular weight (MW) of the analyte or equal to the m/z of the analyte. Some cases were observed in which the sample was ionized by losing one hydrogen atom, resulting in a (m/z) of minus one for the detected species. A high flow rotary pump is employed to reduce the pressure inside the ionization chamber and to remove species that undergo non-reactive collisions with oxonium ions. An ion funnel of stacked ring electrodes with decreasing inner diameter collects ions from the ionization source to avoid free-jet expansion, confines and transmits the ion beam to the RF lines. A schematic of the inlet interface and ionization chamber are shown in Figure 1.

Exiting the ion funnel, the ion beam proceeds to the transfer line where a set of ion guides (octapole, quadrupole, and hexapole), a high-vacuum lens and a beam forming slit focus and transfer the ion beam to the ToF section. A differential aperture is located after the funnel and the produced ions are gated into the octapole at a pressure of 5x10-3 mbar. Turbo pumps are deployed along this line to successively reduce the pressure up to 10-7 mbar entering the ToF analyzer. Different RF amplitude and frequency settings can modify the mass-to-charge ratio cut-off and maximize the transmission of both the low and high m/z ranges; therefore, the desired m/z range needs to be decided beforehand and the RF amplitude and frequency settings can be adjusted accordingly. Passing from the RF line to the ToF, there is a thermalization period of 20 ms achieved by collisions with the carrier gas molecules.

The ToF part is an orthogonal accelerator in which the ion beam follows a V-shaped path at the end of which lies the detector. Electric fields are used to accelerate the ions in different time pulses. Depending on the time needed for a batch of ions to cross



the ToF section after high-frequency and high-voltage pulses are applied, their m/z can be defined. Different time delays can be set and the overall resulting range of the m/z is 30-5000 Th. The ToF mass analyzer is tuned to second-order time focusing

and the analog signal is sampled by a 14bit, 2 Gs s⁻¹ ADC. Suitable software has been developed to control the different instrument settings and display the spectra acquired.

The detection limits of the novel charge transfer oToF-MS fall below 2 ppt. In Kaltsonoudis et al. (2023) a more detailed analysis of the detection limits can be found. The mass resolving power (M/dM) at 500 Th of the oToF mass analyzer is 20,000 (fwhm). Fragmentation of higher molecular weight compounds was detected during chamber experiments and the observed

fragmentation patterns are described in Kaltsonoudis et al. (2023). The electric field is 89 Td (10-17 V cm$^2$).

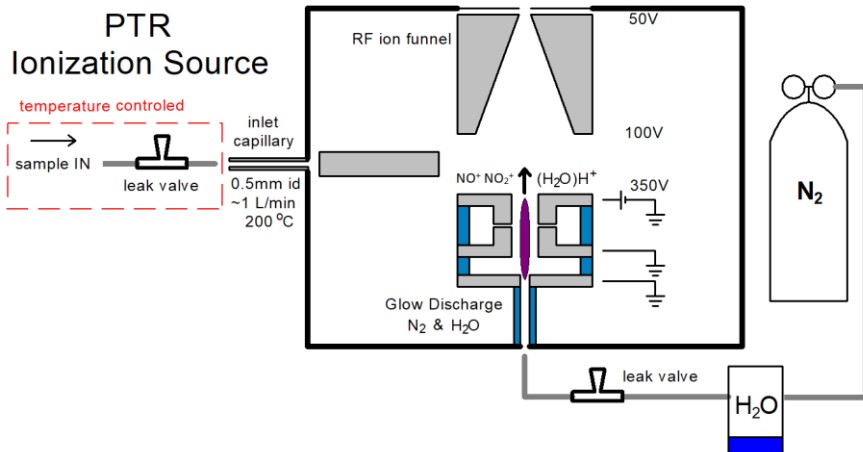

**Figure 1. Schematic of the charge transfer oToF-MS ionization source.**

The instrument was operated at the DEM station during May-August 2023. The site, located in the suburbs of Athens, is part of the Global Atmosphere Watch (GAW) program, part of the Aerosol, Cloud, and Trace Gases Research Infrastructure

(ACTRIS) and the PANhellenic infrastructure for Atmospheric Composition and climate change (PANACEA) (37.995o N, 28.816o E). The site is at 8 km from the city centre, at the foothills of mountain Hymettus, and is influenced by both anthropogenic emissions from the city and biogenic VOCs from the trees in the mountain (Figure 2). The area is also affected by traffic emissions from a nearby highway. These factors make the area a challenging environment for VOC monitoring and source identification. The instrument was operated with an inlet flow rate of 0.5 L min-1 and the ionization chamber pressure

ranged between 2.2 and 2.4 mbar.





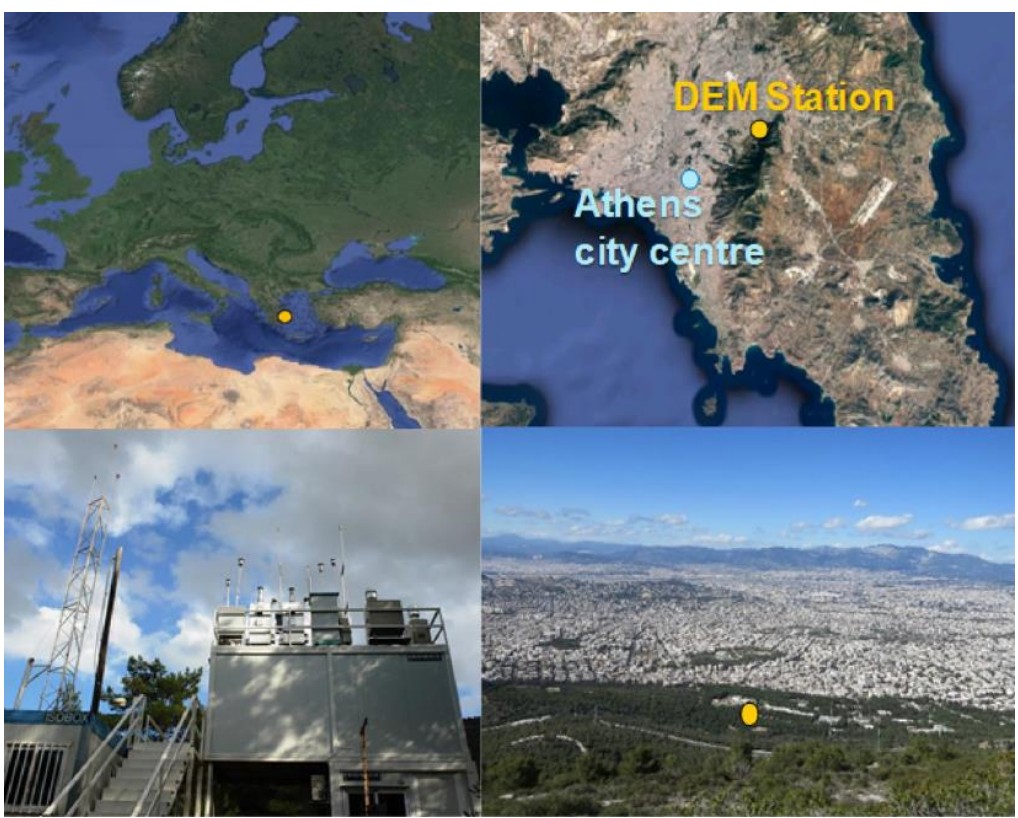

**Figure 2.The Demokritos Atmospheric Aerosol Measurement station in Ag. Paraskevi, Athens, Greece. The maps were obtained from © Google Maps (http://maps.google.com, last access: 19 January 2023) imagery 2021 Terrametrics, Mapdata 2021 and modified by the authors.**

## 2.2 Additional instrumentation

Concurrent data for the non-refractory PM1components were also available from a Time-of-Flight Aerosol Chemical Speciation Monitor (ToF-ACSM). An AE33 aethalometer provided measurements of the equivalent black carbon concentrations, and NOx and O3 concentrations were measured by the respective monitors. The collocation of the instrumentation above is used in order to compare and verify the findings of the targeted VOC analytes with respect to the pollution sources influencing the DEM station.

Additional analysis such as positive matrix factorization (PMF) of the particulate species as well as conditional probability function (CPF) analysis of the wind speed and direction in respect to the detected VOC species was also performed.

### 2.2.1 ToF-ACSM

The ToF-ACSM by Aerodyne Research Inc. (Billerica, MA, USA) was deployed at the station during August 2023 for contemporaneous chemical characterization of $PM_1$ particles and discrimination between organic species and inorganic ions, such as sulphate, nitrate, ammonium, and chloride. The operational principle of the instrument is described in detail in Fröhlich



et al. (2013). The instrument was coupled with a Nafion dryer to avoid high relative humidity of the sampling line. The time resolution of the measurements was 10 min, which were then averaged to 30 min. The relative ionization efficiencies were determined after calibration and set to 3.37 for ammonium and 0.77 for sulphate, while the ionization efficiency was 179.9
ions pg[-1]. A composition dependent collection efficiency was applied to the concentration of the species, following the recommendations by Middlebrook et al. (2012).

### 2.2.2 Aethalometer

An AE33 Aethalometer (Magee Scientific Corp., Berkeley, CA 94703, USA) provided the values of light absorption coefficients and equivalent black carbon concentrations (*eBC*) (Drinovec et al., 2015). In this study the wavelength at λ= 880
nm was used and the conversion of absorption coefficient to concentration was made using a MAC value of 4.6 m$^2$ g$^{-1}$ (Kalogridis et al., 2018). The absorption Ångström exponents used for distinction between pure fossil fuel combustion ($\alpha_{ff}$) and pure biomass burning aerosol ($\alpha_{bb}$) were 0.9 and 2.0 respectively (Diapouli et al., 2017a). The respective fractions of *eBC* are denoted as *eBC$_{ff}$* (black carbon from fossil fuel) and *eBC$_{bb}$* (biomass burning-related black carbon).

### 2.2.3 NO$_x$ and O$_3$ analyzers

Analyzers from the air quality monitoring station of the Greek Ministry of Environment and Energy network were measuring simultaneously inside the NCSR Demokritos campus, providing with 1h time resolution NO$_x$ and O$_3$ concentration measurements.

### 2.2.4 Meteorological data

The meteorological data, including temperature, solar radiation, and wind speed and direction were obtained from sensors
installed on a meteorological mast, at 10 m above the ground.

### 2.3 Data analysis

### 2.3.1 Positive Matrix Factorization

The sources of organic particulate matter, as measured by the ToF-ACSM were determined by applying Positive Matrix Factorization (PMF). In PMF, the data matrix (**X**) is described as the product of the time series of each factor (**G**) and the
factors' profile (**F**), plus a residual matrix (**E**):

$$\mathbf{X} = \mathbf{F} \, \mathbf{G} + \mathbf{E}, \tag{1}$$

The aim of the PMF model is to minimize the error metric Q, which is defined the sum of the squares of the ratio of the residuals (e) to the uncertainties (σ) of each point in the **X** matrix data:

$$Q^m = \sum_{i=1}^{m} \sum_{j=1}^{n} \left(\frac{e_{ij}}{\sigma_{ij}}\right)^2, \tag{2}$$



The a-value approach was used to constrain the profiles of primary factors in order to avoid the rotational ambiguity problem that can lead to a high number of solutions from the rotation of the corresponding vectors. Moreover, the resampling technique (bootstrap) was employed to estimate the uncertainty of the PMF solutions (Efron et al., 2000). The software used for the deconvolution of the organic sources was the Source Finder (SoFi) Pro (Datalystica Ltd, Villigen, Switzerland) (Canonaco et al., 2013) that uses the multilinear engine (ME-2) as a solver (Paatero, 1999). At first the model was run without any constraints and the factors identified in previous years at the same station as described in Zografou et al (2022) were retrieved. The final solution was reached after performing 100 simulations enabling the bootstrap technique and constraining the 3 primary profiles with random a-values from 0 to 0.5 (allowing for maximum 50% variability in each profile). The three primary factors were: a hydrocarbon-related organic aerosol (HOA) factor linked mainly to traffic emissions, one cooking OA (COA) and a factor associated with biomass burning emissions (BBOA). The other two factors represented secondary aerosols: a more oxidized (MO-OOA) and a less oxidized oxygenated OA (LO-OOA) factor.

### 2.3.2 Wind analysis

The Conditional Probability Function (CPF) plots were produced through the OpenAir package in R studio (Carslaw and Ropkins, 2012) using the polar plot function. The bivariate polar plots of concentrations were used in order to identify the most potential origin of the VOCs in spatial terms. The CPF determines the probability that the concentration of an ambient compound is higher than a value (in this study the mean) in a specific wind direction and speed.

## 3 Results and discussion

### 3.1 Identification and quantification of VOCs

The instrument's operational stability, allowing continuous measurements of specific VOCs from May to September 2023, indicates its reliability for field deployment. One of the major advantages of the prototype instrument is its high-resolution, with mass resolving power of 20,000 (fwhm) at 500 Th, enabling the differentiation between compounds with subtle molecular differences. Moreover, the instrument's sensitivity is a key to its analytical power, with detection limits below 2 ppt, as demonstrated by the calibration using a multi-component VOC standard (Kaltsonoudis et al., 2023), permitting the detection and quantification of trace levels of VOCs, essential for understanding their dynamics in ambient environments.

The first deployment of the prototype oToF-MS for field measurements resulted in the identification of various VOCs in the suburban environment of Athens, including both anthropogenic and biogenic compounds. The present study focuses on selected compounds, namely acetone, isoprene, benzene, toluene, and xylene, and on the analysis of their temporal variation. The oToF-MS was calibrated for the respective VOCs at the DEM station using a standard mixture, following Kaltsonoudis et al. (2023). The campaign average mass spectra of these VOCs are displayed in Fig. S1.

An external calibration needed to take place due to a shift of the mass to charge ratios by a standard value when the instrument was deployed for field measurements. According to the time-of-flight principle, the mass of an ion is given by Eq. (3):



$$m = \frac{2E}{d^2} t^2 , \tag{3}$$

where $m$ is the ion mass, $E$ is the energy given at an ion for acceleration, $d$ is the distance the ion has to cover from the ion pulser to the detector (the time-of-flight tube length) and $t$ is the respective time needed. Given that the power provided to the ions was constant, the reason behind this mass shift is probably a temperature-related expansion of the flight tube due to the different sampling conditions in the field and especially temperature variations that can cause expansion or contraction of the ToF column.

The relative abundance of the VOC fragments was similar but not identical to that reported in Kaltsonoudis et al. (2023). The respective relative abundancies at each $m/z$ in the previous study and the current one are shown in Table 1. As discussed in Kaltsonoudis et al. (2023), both electron ionization and proton transfer have been observed to take place at the ionization source. A similar pattern is observed in the field deployment as in the lab experiments, while the difference observed in the relative abundancies of some $m/z$ values is attributed to the different sampling conditions, since the fragmentation pattern may

be condition-specific for each compound.




**Table 1. Relative abundance of the m/z of each compound from the previous study in lab settings and the current field study.**

| Compound | m/z | Relative abundance- Kaltsonoudis et al. (2023) | Relative abundance- This study |
|---|---|---|---|





| | | | |
|---|---|---|---|
| Acetone | 59 | 100 | 100 |
| | 60 | 2 | 4.3 |
| Isoprene | 67 | 30 | 22 |
| | 68 | 99 | 43 |
| | 69 | 100 | 100 |
| Benzene | 78 | 100 | 100 |
| | 79 | 16.5 | 49 |
| Toluene | 92 | 100 | 100 |
| | 93 | 43 | 58 |
| Xylene | 106 | 100 | 100 |
| | 107 | 45 | 67 |
| | 108 | 26 | - |


Acetone was detected in the protonated form $(C_3H_6O+H)^+$, with $m/z$ 59 as the prevalent peak and the second peak at $m/z$ 60 at 4.3% close to the theoretical one (3.2%) and slightly higher than in the previous study by Kaltsonoudis et al (2023) at 2%. Isoprene was detected primarily in the protonated from $(C_5H_8+H)^+$ at $m/z$ 69, but also in the non-protonated form $(C_5H_8)^+$ at 43% (which was reported at 99 % in Kaltsonoudis et al. (2023)) and finally in the form of hydrogen subtraction $(C_5H_8-H)^+$ at

$m/z$ 67. Benzene was in both studies dominated by the non-protonated peak at $m/z$ 78, but the higher relative abundance at $m/z$ 79 (49% in this study, over 6.5% in theory) is an indication of contemporaneous protonation of benzene. A higher efficiency of protonation is observed (36.5%) compared to our previous study (10%). Toluene also appears mainly in the non-protonated form at $m/z$ 92 and at a higher percentage in the protonated form $(C_7H_8+H)^+$ (58%) than theoretically (7.6%) and different than the previous study, indicating proton transfer with higher efficiency (50%). Finally, xylene was detected at 2 ion peaks, the

non-protonated at $m/z$ 106 and the protonated but at higher relative abundance than theoretically (67% in contrast to 8.7% theoretically) indicating almost 60% efficiency of the protonation, while in the previous study it was represented by the 3 peaks shown in Table 1.

Figure 3 presents the time series of these VOCs during the summer campaign with an averaged time resolution of 1 h.



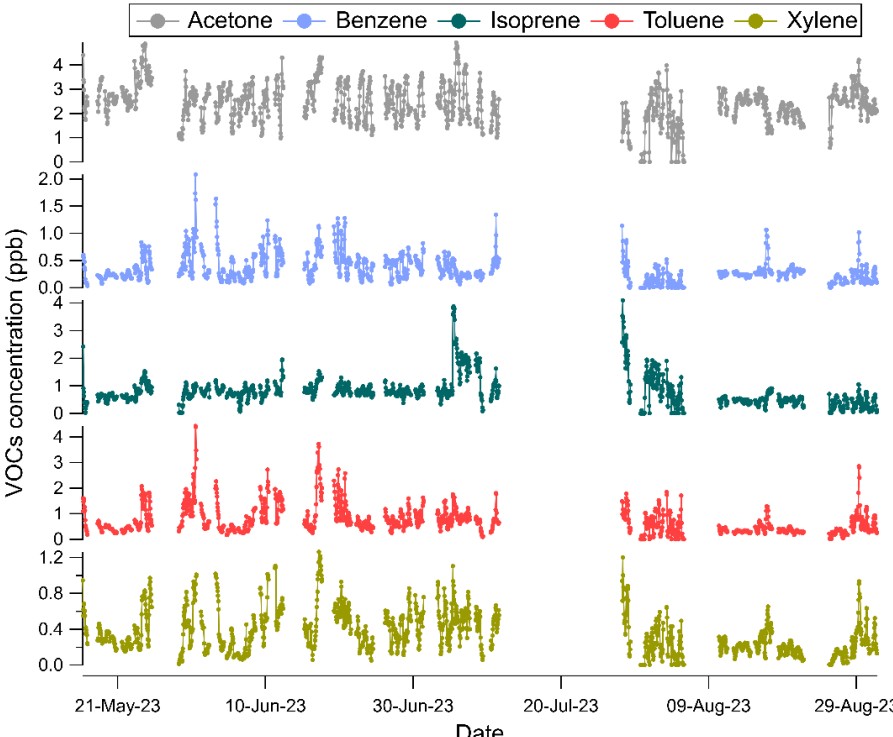

**Figure 3. Temporal variability of the selected VOCs (acetone, benzene, isoprene, MACR, toluene and xylene) measured by the prototype charge transfer oToF-MS.**

Acetone shows the least variability during the four months measurement period, with concentrations varying from 1-4 ppb. Conversely, isoprene shows increased concentration in July, compared to the other months. The aromatics (benzene, toluene, and xylene) had both periods of relatively low level and 7 events of higher concentration levels, 6 of which had similar increases for all three species. In the last event only benzene and toluene increased. Most of these events are accompanied by peaks in *eBC* and/or NO$_x$ concentration pointing out to fresh combustion emissions. The concurrency of the BTX compounds, confirmed also by their R-Pearson correlations shown in Fig. S4 (R$^2$ between 0.82 and 0.89), indicates a high possibility of one source of aromatics, namely vehicle emissions. The concentration distribution of all compounds is shown in Fig. 4. Additional information about the distribution of the concentration measurements can be found in the SI.



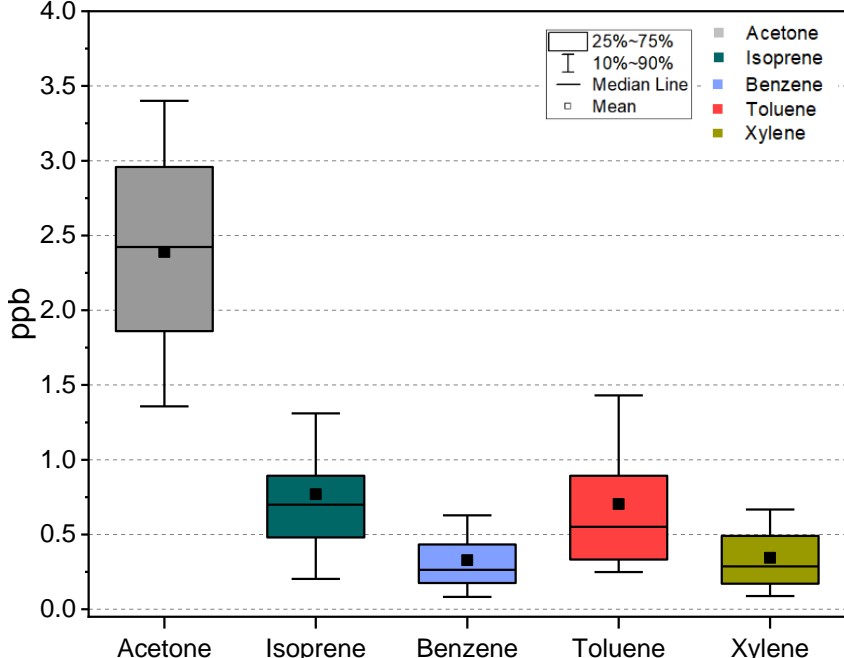

**Figure 4. Concentration box plots of the main VOCs measured with the new charge transfer oToF-MS system.**

Comparison with previous studies in this site during summer (Kaltsonoudis et al., 2016), yielded significant agreement in the concentration levels for the selected VOCs. More specifically, toluene and benzene had average values of 0.81 ppb and 0.22 ppb in Kaltsonoudis et al (2016), compared to the reported at 0.7 ppb and 0.3 ppb in this study. The toluene to benzene ratio was equal to 3.4 for the summer campaign in Kaltsonoudis et al. (2016), while in the present study is equal to 2.3, both typical values for traffic-influenced measurements (Vardoulakis et al., 2002; Buczynska et al., 2009). Isoprene, at m/z 69, which is a biogenic compound, with increased concentration in the summer months, was reported at 0.73 ppb during the summer of 2012 and here had an average level of 0.8 ppb. A higher concentration was reported in the study of Kaltsonoudis et al. (2016) for xylene and acetone compared to this study (0.67 ppb and 4.28 ppb, respectively, compared to 0.3 ppb and 2.4 ppb in this study). In both studies acetone was the most abundant compound.

**3.2 Links with concentration of other pollutants**

The measured concentrations of the rest of the pollutants are shown in the SI (Fig. S2). The *eBC* concentration is rather stable during the campaign, with some pronounced peaks after July 20, attributed to wildfires, which however were not captured by the oToF-MS which was not operated at the time. $eBC_{bb}$ is generally significantly lower than $eBC_{ff}$, especially during the summer months, when vehicle emissions are relatively high, but there is no residential biomass burning for heating purposes. $NO_x$ and $O_3$ do not present any strong peaks during the measurement period. Both *eBC* fractions, as well as $NO_x$ and $O_3$ had a



period with decreased concentrations during 18-20 July, which again could not be compared to VOCs levels, since the instrument was not operated at the time.

Figure S4 shows the profiles, and the temporal and diurnal variation of the organic aerosol factors from the ToF-ACSM, as estimated by the PMF. PMF could explain the ACSM measurements with five factors. The three primary factors included HOA, a factor related to fossil fuel combustion mainly from vehicle emissions, COA, a cooking-related factor, and BBOA, OA emitted by biomass burning. The two oxygenated OA factors represented more and less oxidized (MO-OOA and LO-OOA, respectively), mainly based on the signal at $m/z$ 44. The diurnal variation of the factors also shows the expected trends, with the HOA average diurnal profile showing two peaks during the day, coinciding with rush hours and COA presenting a noon and an evening peak. BBOA had a peak in the morning which was related to its concentration peaks due to the wildfires in mid- and late- August affecting the site. The two OOA factor exhibit rather stable behaviour during the day as they are due to a large extent to long-range transport.

The diurnal variability of the VOCs was studied together with the respective variability of other pollutants with potentially common emission sources. Pollutants with similar average diurnal patterns are depicted in Fig. 5 and the corresponding correlation are shown in Table 2.



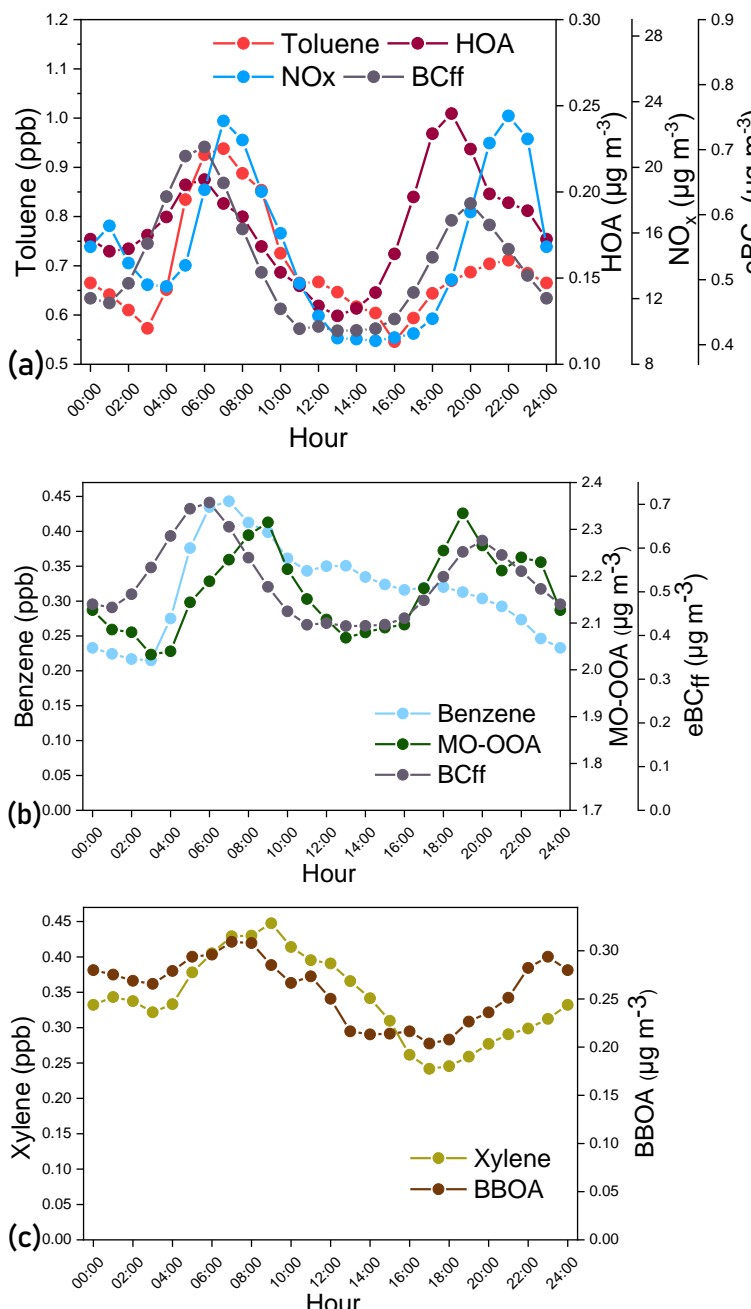



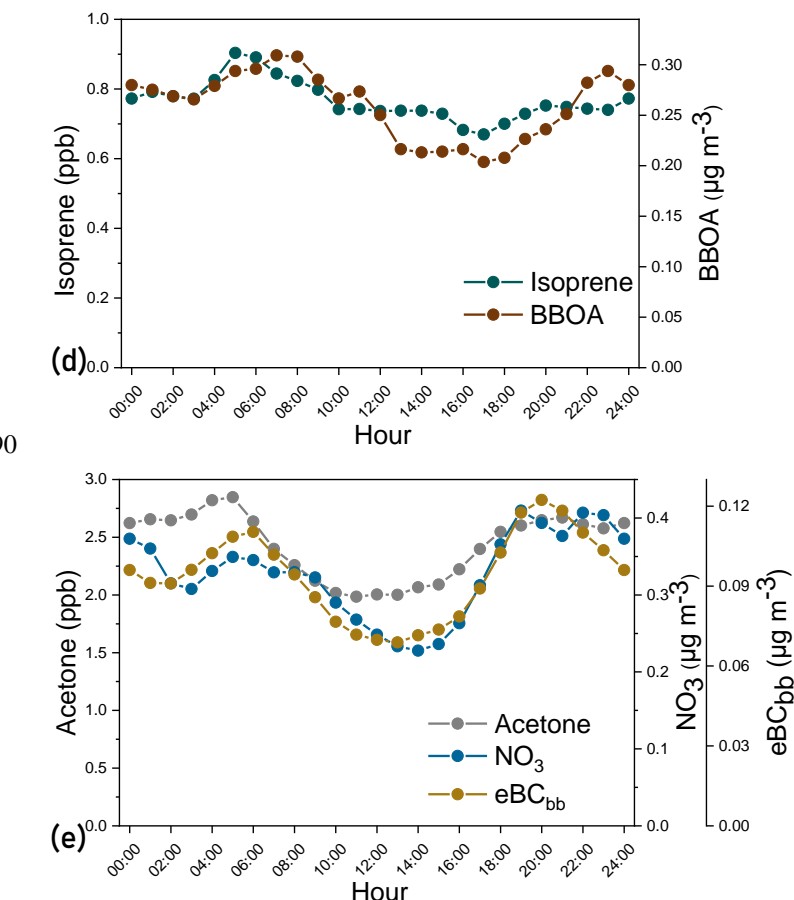


**Figure 5.** Average diurnal concentration variation of the selected VOCs and of pollutants with similar behaviour.






**Table 2. R-Pearson coefficients of the investigated VOCs with other pollutants at a resolution of 1 h.**

| Compound | External tracer | R-Pearson |
|---|---|---|
| Toluene | HOA | 0.30 |
|  | $eBC_{ff}$ | 0.64 |
|  | $NO_x$ | 0.67 |
| Benzene | $eBC_{ff}$ | 0.31 |
|  | MO-OOA | 0.42 |
| Xylene | BBOA | 0.66 |
|  | $NO_x$ | 0.34 |
| Acetone | $NO_3$ | 0.77 |
|  | $eBC_{bb}$ | 0.84 |
|  | Cl | 0.53 |
| Isoprene | BBOA | 0.77 |

In general, the diurnal variation of toluene and benzene align closely with traffic-related markers, demonstrating two characteristic peaks during rush hours. Toluene correlates with $NO_x$ (R=0.67) and $eBC_{ff}$ (R=0.64). Benzene exhibits medium
correlation with $eBC_{ff}$ (R=0.31). During the morning rush hours (starting at 08:00 LT), the concentration of the aromatic VOCs increases. More specifically, toluene reaches on average 1 ppb, while benzene and xylene go up to 0.5 ppb. The behaviour during the evening rush hour is similar, but the toluene increase is a little less than in the morning. Xylene, although correlated with toluene and benzene, is also emitted by biomass burning (Wang et al., 2014), something supported in this case by its correlation with BBOA. Biomass burning episodes, which in the case of summer in Greece are related to wildfires. The
complete time series of the aromatic compounds and pollutants ($eBC_{ff}$ and HOA) can be found in Fig. S5.

Isoprene also originates from both biogenic and anthropogenic sources (Panopoulou et al., 2020). Isoprene correlated well with BBOA (R=0.77). It has been previously reported that isoprene can be emitted by tree combustion (Pallozzi et al., 2018). In Fig. 5 the average diurnal profile of isoprene presents an early morning peak probably related to the increasing biogenic emissions combined with the relatively low mixing height at this period.

Acetone can originate from primary and secondary sources, both biogenic and anthropogenic in nature. Its average profile is characterized by two peaks during the day, coinciding with those of particulate nitrate from and also $eBC_{bb}$. Holzinger et al. (2005) reported that biomass burning is a significant source of acetone.



The diurnal variation of the aromatics displayed the same patterns as those reported by Kaltsonoudis et al. (2016) for the summer 2012 campaign at the same site. Since the aromatic compounds are mainly due to the vehicle emissions this consistent

diurnal variability is encouraging for the performance of the ToF-MS.

## 3.3 Geographical origin of VOCs

Additional insights about the source areas of these VOCs can be gained relating their concentration levels with the corresponding wind direction and speed. For this reason, the OpenAir package from the R studio was utilized. Figure 6 depicts these plots for each VOC. Please note that the centre of Athens is located to the west-northwest of the sampling site.

Benzene and toluene predominantly originate from the city centre, where traffic emissions are more prominent. Xylene appears to have two major emission source areas, showing strong influence by emissions from the city centre, but also from emissions to the east and south east of the station, which could be related to one of Attica's ports, located at the East of DEM station.

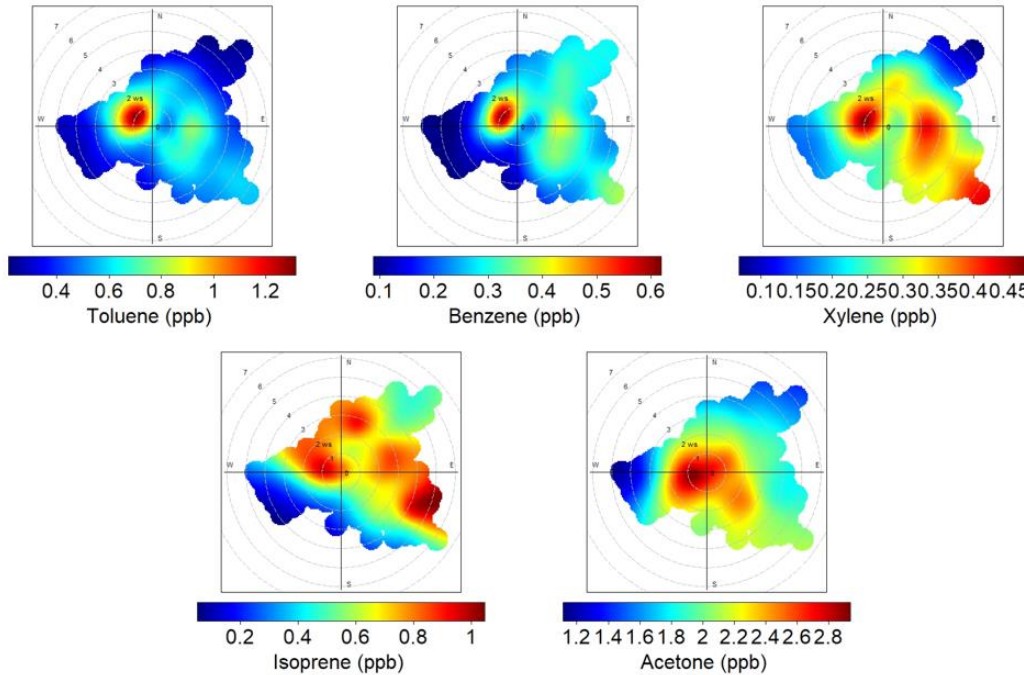

**Figure 6. Polar plots of the selected VOCs (75th percentile): a) toluene, b) benzene, c) xylene, d) acetone and e) isoprene.**

Acetone, related to both anthropogenic and biogenic sources, is shown to have an emission hotspot at the centre of the city, related to anthropogenic emissions, but also from the mountainous region close to the station. Isoprene's broader source area also supports its biogenic and anthropogenic nature, originating from both city-related emissions and the nearby mountainous region. Further insights can be gained by examining the spatial distribution of isoprene sources for each month separately (Fig. S6). In July the isoprene emissions from the city centre are not that important. However, this month was characterized by the

highest temperatures during the measurement period, thus higher biogenic isoprene emissions are expected. The higher

concentrations of isoprene in July compared to the other months can lead to the conclusion that biogenic isoprene is more important than anthropogenic isoprene.

## 4 Conclusions

This study presents the results of the first field deployment of a novel charge transfer oToF-MS instrument for real-time detection of VOCs at a suburban station in Athens from May to August 2023. Focusing on measuring five ambient VOCs (toluene, benzene, xylene, isoprene and acetone), the successful implementation was confirmed by comparison of the VOCs levels with previous studies at the same station, and by their correlation with compounds with common emission sources measured by collocated instruments.

The high temporal correlation of the aromatic compounds (toluene, benzene and xylene) indicated that vehicle emissions is a major source. This was also confirmed by their significant correlation with other traffic markers, such as $NO_x$, *eBC$_{ff}$* and HOA. Isoprene had higher levels in July, when emissions are expected to increase due to higher temperatures. It concentration was higher when air masses arrived at the site from the nearby forest.

Future work includes improvement of the instrument's software to facilitate long-term measurements with minimum user intervention. Additionally, in terms of data analysis, identifying the sources of the VOCs in this environment by applying source apportionment models will retrieve the current VOCs sources and their evolution in time and link them with previous studies.

*Author Contribution*

OZ, MG, KE, DP, AL, EP, EK, CK and SNP organized the instrument deployment. OZ conducted the measurements, and analyzed the data with input from CK, AL, AM, and DP. OZ wrote the original manuscript with inputs from CK, SNP, DP, and EK. All authors discussed, reviewed and edited the manuscript.

*Competing Interests*

The authors declare that they have no conflict of interest.

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
