# Peer review of "Field evaluation of a novel charge transfer ionization TOF MS for ambient VOC measurements"

_EGUsphere, 2024_

## Author Comment (AC1)

**Reviewer #1:**

The paper titled "Field evaluation of a novel charge transfer ionization TOF MS for ambient VOC measurements" discusses ambient measurements results from a newly designed and built TOF MS that was deployed in Athens. The instrument details are primarily discussed in a previous paper and here the authors focus on validating the instrument for field measurements. For this purpose, other traditional instruments were co-located at the measurement site with which the results are compared. The paper specifically focuses on 5 VOCs that generally have well-defined anthropogenic or biogenic source contributors.

It is straightforward to read, which is appreciable. However, there are several grammatical errors across the manuscript that suggests the writing has not been thoroughly vetted by the authors. I would highly recommend that the authors review their writing more closely to eliminate such errors. For e.g., "sear" in line 24, confusing use of "in" vs. "at" vs. "of" at several instances etc. Periods are missing at the end of some sentences.

*The authors tried to correct any grammatical errors.*

I also felt that the introduction builds very limited ground for the need to develop a new mass spectrometer. There should be some more discussion of what already exists, what is it that is missing and why this mass spectrometer is important. Perhaps this is discussed in the previous instrument-focused paper but some discussion is necessary here as well in my perspective since measurements from a new instruments are discussed.

*We thank the reviewer for the comment. The introduction was revised to include more discussion on the importance of the development of such an instrument. The added sentences are in Lines 47-56 as follows:*

*"The present instrument was previously characterized in Kaltsonoudis et al (2023), where under laboratory conditions the results showed extremely lower detection limits compared to existing PTR-MSs, and also the ability to detect VOCs at higher m/z ranges, although with the constraint of focusing on specific size ranges each time, thereby losing lower m/z values when the focus is on higher values. Real-time instrumentation for VOCs measurements is essential in order to obtain a picture of the outdoor and indoor air quality, and is less risky compared to offline analyses, where sampling can cause many artefacts (reactivity of VOCs with each other or with other material, accumulation of water etc). Very low detection limits are required for atmospheric measurements of VOCs, since many VOCs appear in very low concentrations. Apart from impressive detection limits, this novel charge transfer MS also presents competitive features, such as extraordinary mass resolution (15000 M/dM at m/z 250). Moreover, a high range from VOCs to IVOCs can be measured."*

Line 187: It is not clear what an "external calibration" means. Please provide more detail. I understand that ambient temperature fluctuations can have an effect on the flight path length but how was the instrument externally calibrated?

*The reviewer is correct, the term external calibration needed more clarification. The authors meant to explain that reassignment of the m/zs detected by the instrument took place after obtaining the data.*

*The respective part of the manuscript was revised accordingly:*

*Line 212: "A post-measurements external calibration needed to take place due to a shift of the mass to charge ratios by a standard value when the instrument was deployed for field measurements."*

Line 197: "electron ionization" can be easily confused with EI. I think the authors mean charge transfer by dropping an electron. Please revise for clarity.

*The reviewer is correct; the authors were referring to charge transfer by removal of one electron. The respective section of the manuscript was corrected.*

Line 199-200: The authors state that fragmentation pattern may be condition specific. Do they mean instrument-conditions-specific or ambient conditions-specific such as the influence of relative humidity? If ambient, please explain how providing some citations.

*We thank the reviewer for the comment. Different fragmentation can both occur due to different settings on the voltages of the funnel, lenses, ionization source etc., as well as the ambient conditions (such as RH). The instrument-specific-conditions were kept stable and the fragmentation was checked through the calibration process.*

I have several questions about the data shown in figure 5:

Figure 5b: Why does MO-OOA follow BCff trends in the second half of the day and BCff does not correlate with benzene? Also benzene does not show a trend that would be consistent with evening commute hours. How can this be explained?

*Enhanced photochemistry in the summer results in quick formation of oxidized OA, such as MO-OOA from VOCs (through gas-phase oxidation and gas-particle partitioning), resulting in Secondary Organic Aerosols (SOAs) presenting a diurnal pattern that could resemble that of primary emissions.*

*Concerning the diurnal pattern of benzene, its less enhanced evening mass concentration during rush hour is in accordance with Wagner et al (2014), explained by more important dilution of benzene that prevails over its increased concentration due to traffic emissions. Therefore, the authors decided to remove external data from the respective plot of benzene diurnal trend.*

Figures 5d/e: BBOA and eBCbb don't seem to line up. BBOA peaks close to mid-night but eBCbb peaks around 8 PM. Acetone appears to level out in the evening irrespective of variations in eBCbb and BBOA. Please explain.

*It's common for BBOA and eBCbb to exhibit weaker correlation during summer, since background levels are measured in the summer when no constant biomass burning source is expected.*

Line 308: Benzene is stated to align with traffic-related markers. However, when Fig 5a/b are compared, HOA and benzene don't align after an initial morning period.

*Again, as stated in the comment concerning Fig. 5b, benzene was shown to align with HOA and eBCff in the morning, but as enhanced dilution of benzene takes place (compared e.g. to toluene), less pronounced evening rush hour peak is observed.*

Line 309-310: I am not sure if I would call R = 0.31 a medium correlation. I'd suggest this to be a minor to poor correlation even considering field situations. This also raises the question of why benzene and eBCff won't align, especially since the authors attribute aromatic species to be mainly emitting from vehicular emissions in line 324.

*The reviewer raises a concern that was previously addressed. The respective part of the manuscript was modified, since an R-Pearson of 0.31 indicated indeed poor correlation. The diurnal variability of benzene was closely related to atmospheric dilution, rather than to its source emissions, which have been identified previously (Kaltsonoudis et al., 2016). Again, this aligns with Wagner et al (2014).*

Lines 318-319: Isoprene's morning peak is attributed to biogenic emissions. However, such enhancement can also be seen in BBOA. How do these things reconcile?

*As mentioned in the manuscript, isoprene can originate from both anthropogenic and natural emissions (Line 316). In July, isoprene was seen to mainly originate from natural emissions by vegetation (supported also by Fig. S6 when no relationship between increased isoprene concentration and wind direction favoring air masses from the center of Athens was found), and the diurnal plot of isoprene followed $O_3$, driven mainly by photochemistry (Fig. S7). On the other hand, in August, isoprene's diurnal variation was shown to match BBOA's, thus connected also with anthropogenic emissions.*

Table 2: I would recommend showing correlations of each target compound with all external tracers instead of a select few to get a better of sense of agreements vs contrasts.

*The authors believe that additional information on this table is not to add any valuable information, since it would include compounds that share no similar emissions with respective external tracers. The correlations already reported are the 'highest' that were observed between the different species.*

Figure 6: I suggest clearly marking the city center on each plot for readers' convenience. The spatial scale is also not clear on these plots. As a reader, I am not sure how far am I looking at the terrain in these plots for source contributions.

*We thank the reviewer for the observation. The city center was marked in the CPF plot of toluene for convenience of the reader.*

Figure S6 shows very interesting distributions for isoprene around Athens during May-August. Isoprene is very high near the city center (from what I understand where the city center is from these plots) in June but low in the mountainous regions in the south-east. The city center is somewhat unchanged in July but a high intensity area appears in the north-east in July with emissions up to 2.5 ppb. This exceeds the 1 ppb max scale when the mountainous sources dominate isoprene contributions in August. I am curious why the sources of isoprene are changing so dramatically over a short span of a couple of months. Are there multiple biogenic sources of isoprene all around Athens other than the mountains in the south-east and north-west? Or the wind direction is flipping around? The authors discuss this briefly in lines 336-342. Perhaps it might be useful to include a wind rose plot in the SI.

*The wind roses for each month were added in the Supplement (Fig. S8). In general June and July seem to be quite similar in terms of wind direction though July has somewhat higher wind speeds. Also May and August wind direction frequencies are quite similar. While there are similarities and differences in the wind patterns for each month, the variability in the isoprene concentrations can also be affected by a number of factors related to the biogenic sources, including ambient temperature, rainfall (or rather the lack of rainfall) and the geographical terrain of the Athens basin which is affected from other areas (more forested areas) of Greece depending on the wind trajectories.*

Minor comment:

I suggest adding ionization reaction equations to section 2.1 to help make the details more comprehensive.

*The reviewer has raised a good point. The ionization reaction equations as also shown in Kaltsonoudis et al (2023) are now included in section 2.1.*

*References:*

Kaltsonoudis, C., Kostenidou, E., Florou, K., Psichoudaki, M., and Pandis, S. N.: Temporal variability and sources of VOCs in urban areas of the eastern Mediterranean, Atmos. Chem. Phys., 16, 14825–14842, https://doi.org/10.5194/acp-16-14825-2016, 2016.

Wagner, P. and Kuttler, W.: Biogenic and anthropogenic isoprene in the near-surface urban atmosphere — A case study in Essen, Germany, Science of The Total Environment, 475, 104–115, https://doi.org/10.1016/j.scitotenv.2013.12.026, 2014.

Kaltsonoudis, C., Zografou, O., Matrali, A., Panagiotopoulos, E., Lekkas, A., Kosmopoulou, M., Papanastasiou, D., Eleftheriadis, K., and Pandis, S. N.: Measurement of Atmospheric Volatile and

Intermediate Volatility Organic Compounds: Development of a New Time-of-Flight Mass Spectrometer, Atmosphere, 14, 336, https://doi.org/10.3390/atmos14020336, 2023.

---

## Author Comment (AC2)

*This manuscript reports on the long-term deployment of a newly developed charge transfer orthogonal ToF-MS (oToF-MS) in Athens, Greece, for online VOC measurements. My major concern is with the scope of the manuscript. The introduction gives the impression that the primary focus of this study is to evaluate the long-term deployment of this new instrument. To achieve this goal, it is essential to compare the measurements to an established method, such as GC-MS or GC-FID. Unfortunately, this comparison is missing in the manuscript. The "Results and Discussion" section extensively discusses the sources and geographical origins of VOCs, which deviates from the main focus. If source apportionment of VOCs is indeed the primary focus, the analysis presented here is superficial, and several conclusions are not well supported. Therefore, my overall recommendation is to better define the scope, include the necessary measurements to support it, and remove the distracting analysis. I do not want to discourage the authors. The development of a highly sensitive instrument with ~20,000 mass resolution is very impressive. I hope to see more detailed characterization, evaluation, and results from this instrument in the future.*

The authors appreciate the concerns raised by the reviewer, but would like to point out that a prior study has evaluated the qualitative and quantitative performance of this novel charge transfer MS using a standard mixture (Kaltsonoudis et al., 2023). The scope of the current study is to assess the instrument's performance under real ambient conditions during field measurements. The authors aimed at deploying the instrument in-situ at a running station and in parallel with other instrumentation measuring atmospheric aerosol properties, chemical characteristics and environmental parameters.

This work evaluates the ability of the novel instrument to perform unattended real-time, online, in-situ measurements. Since the authors agree that a comparison as the one the reviewer suggested will be indeed interesting as a next step, the following was added in the conclusions section of the manuscript:

Line 373: "A complete evaluation is pending for the instrument, where comparison of the field measurements to an established method, such as GC-MS or GC-FID, will have to be performed."

Nevertheless, the authors would like the reviewer and the editor to consider this manuscript for publication following the revisions, given that the results of this study support the successful implementation of this novel instrument. This is a preliminary study presenting the first deployment of this instrument that highlights its impressive capabilities, including high sensitivity and low detection limits. Therefore the authors feel that a study demonstrating its successful four-month in-situ deployment would be a valuable contribution to the scientific community.

**Other comments:**

1. *Line 39: It is true that proton transfer causes less fragmentation than electron impact ionization. However, the strong electric field in the PTR still causes significant fragmentation, which introduces challenges in product identification. A recent study by Coggon et al.1 showed that fragmentation from higher-carbon aldehydes and cycloalkanes substantially contributes to m/z 69 and interferes with isoprene measurements.*

   The reviewer raises an interesting concern that falls however outside of the scope of this study. Whether interferences in the isoprene signal caused by higher-carbon aldehydes and cycloalkanes fragmentation took place, this is a topic to be studied in the future, both for PTR-MS systems in general, and for this charge transfer instrument in particular. For now we could state in our manuscript that this recent study has not been taken into account, and therefore there lies the possibility that the measured isoprene signal could contain interferences.

2. *Line 71: Heated SS tubing may cause measurement interference. For example, hydroperoxides may convert to carbonyls on metal surfaces2. Again, this is why validation of VOC measurements by an independent instrument should be included in this study.*

   The reviewer's point here needs to be taken into account in the future, and could be the focus of another study. As far as our measurements are concerned, the SS tubing length was the lowest possible, in order to avoid such interferences, and we therefore believe that such bias was not introduced in our measurements, but could be the topic of a future study.

3. *Table 1: Are the relative abundances reported in this study from calibration or ambient measurements? As mentioned above, ambient measurements may have interferences.*

   The relative abundancies reported here are from ambient measurements and are compared to laboratory conditions relative abundancies from Kaltsonoudis et al (2023) for the same instrument.

4. *Figure 5: Why is benzene shown in the same plot as MO-OOA, as they do not have a strong correlation?*

   The reviewer is right, MO-OOA was seen to present quite similar diurnal pattern with benzene, but that's only due to poor correlation with other

markers, and since there isn't any correlation expected, MO-OOA was removed from the respective plot.

5. *Lines 346-349: The logic here is problematic. Comparing VOCs and OA factors is useful for understanding their sources. However, given that both have complex sources in the atmosphere, it is challenging to use one measurement to evaluate the reliability of the other. In other words, it is difficult to use "the relationship between VOCs and OA factors" to demonstrate "the successful implementation of the new oToF-MS."*

The reviewer is correct. In this study, following the absence of any instrument that measures VOCs, we used a combination of other measurements to evaluate the instrument, only in the sense that the obtained results are environmentally reasonable. That being said, we used tracers to compare VOCs with similar emission sources, as well as polar plots, to reassure the expected geographical origin of such emission sources, and based on previous studies (Kaltsonoudis et al., 2016). Finally, we compared the overall average values with previous values recorded at the same station, concluding that VOCs levels did not vary much between the years.

Reference

Kaltsonoudis, C., Zografou, O., Matrali, A., Panagiotopoulos, E., Lekkas, A., Kosmopoulou, M., Papanastasiou, D., Eleftheriadis, K., and Pandis, S. N.: Measurement of Atmospheric Volatile and Intermediate Volatility Organic Compounds: Development of a New Time-of-Flight Mass Spectrometer, Atmosphere, 14, 336, https://doi.org/10.3390/atmos14020336, 2023.